# Ecological Stoichiometric Characteristics and Adaptive Strategies of Herbaceous Plants in the Yellow River Delta Wetland, China

**DOI:** 10.3390/biology14091132

**Published:** 2025-08-27

**Authors:** Mengjiao Luo, Jiaxuan Liu, Fanzhu Qu, Bowen Sun, Yang Yu, Bo Guan

**Affiliations:** 1The Institute for Advanced Study of Coastal Ecology, Ludong University, Yantai 264025, China; mjwork_l@163.com (M.L.); 17736959784@163.com (J.L.); wow9421@gmail.com (B.S.); 2College of Agricultural and Environmental Science, University of California, Davis, CA 95616, USA; clkyu@ucdavis.edu

**Keywords:** coastal wetlands, annual and perennial herbs, C:N:P stoichiometry, nutrient allocation, soil salinity, the Yellow River Delta

## Abstract

The growth and survival strategies of plants in coastal saline–alkaline soils is an important research topic. This study investigates how herbaceous plants in the Yellow River Delta Wetland, China, adjust the balance of key nutrients like carbon, nitrogen, and phosphorus in their leaves and stems to adapt to their environment. The research compares annual plants, which grow quickly but are more vulnerable to stress, with perennial plants, which grow slowly but are better at surviving harsh conditions. The study found that annual plants had higher nitrogen and phosphorus levels but lower carbon content, while perennial plants had more carbon and lower nitrogen and phosphorus levels, showing their strategies for surviving in saline-alkaline soil. Leaves contained more nutrients than stems, reflecting how plants prioritize their resources for photosynthesis. Soil factors, particularly phosphorus availability, strongly influenced how plants allocate nutrients, and this variation was more important than plant life form. These findings offer valuable insights into how plants in coastal wetlands adapt and help guide conservation efforts to restore these ecosystems effectively.

## 1. Introduction

Ecological stoichiometry focuses on the equilibrium of multiple chemical elements in ecosystems and recognizes that the proportionality of elements affects the growth, metabolism, and balance of ecological functions of organisms [1,2]. Carbon (C), nitrogen (N), and phosphorus (P), as fundamental elements in plant physiological metabolism, interact synergistically in internal energy-material cycling [3,4,5]. The content and stoichiometric ratios of C, N, and P provide valuable insights into plant growth rates and nutrient limitations [6,7], shedding light on how plants utilize nutrients in various ecological environments and adapt to their surroundings [8,9]. The C:N (R_CN_) and C:P ratios (R_CP_) can be used to assess a plant’s ability to absorb nutrients and assimilate C, while the N:P ratio (R_NP_) is employed to evaluate the nutrient limitation status and N saturation level of an ecosystem [10]. Additionally, the Growth Rate Hypothesis (GRH) posits that R_CN_ and R_CP_ can effectively reflect the health and growth status of plants [11,12,13]. These contribute to the scientific understanding of plant survival strategies.

Plant stoichiometric patterns have been extensively studied across different regions and scales, revealing the intrinsic relationships among plant nutrient elements and their responses to external environmental factors [14,15,16,17]. For instance, a broad biogeographic pattern shows that plant leaf N and P contents increase from the tropics to the cooler and drier midlatitudes, while the R_NP_ increases with mean temperature and toward the equator on a global scale [18]. At regional scales, studies reveal that plants reduce stem biomass allocation under arid and hot conditions as an adaptation to environmental stress [19]. In wetland ecosystems, nutrient limitations and salt stress constrain physiological processes like photosynthesis and nutrient mineralization, while plant nutrient allocation patterns reflect their trade-off strategies under environmental stress [20]. Compared with other wetlands, coastal wetland plants face more intense saline-alkali stress, and their elemental allocation strategies exhibit unique adaptive characteristics.

Coastal wetlands, situated at the ecotone between terrestrial and marine ecosystems, exhibit more complex nutrient cycling dynamics due to intense anthropogenic activities and hydrodynamic effects [21,22,23]. The Yellow River Delta (YRD), as a typical coastal saline–alkaline wetland ecosystem, possesses unique ecological characteristics [24]. Due to seawater intrusion from the Bohai Sea, combined with shallow groundwater tables and intense evaporation, soil salinization has become a major ecological challenge for achieving sustainable utilization of land resources in this region. Halophytic species dominate the flora, coexisting with diverse herbaceous plants that utilize specialized physiological mechanisms to tolerate high salinity, while gradually improving saline-alkaline soils [25]. In the unique saline-alkali environment, it is worth exploring how herbaceous plants exhibit their ecological stoichiometric characteristics and allocate nutrients to adapt to the challenging conditions [26]. To date, there is limited research on plant stoichiometric patterns in wetland ecosystems, particularly in coastal wetlands. The understanding of stoichiometric characteristics and inter-organ coordination mechanisms in coastal wetland plants remains incomplete. Based on the above research background, we propose the following hypotheses: (1) under saline-alkali stress, perennial herbaceous plants enhance stress tolerance by increasing the R_CN_ and R_CP_, while annual plants prioritize maintaining higher N and P contents to support rapid growth; (2) inter-organ nutrient allocation drives plant adaptation to saline-alkali environments. This study aims to validate these hypotheses by analyzing the C, N, and P stoichiometric characteristics of different organs of typical herbaceous plants in the YRD, and achieve the following: (1) investigate the ecological stoichiometric patterns and adaptation strategies of plants with different life forms; and (2) elucidate how nutrient allocation patterns among plant organs influence their life history strategies. The findings will offer a more comprehensive understanding of the environmental adaptation strategies of these plants and provide theoretical support for the sustainable development of saline–alkaline soils and conservation efforts of coastal wetlands.

## 2. Materials and Methods

### 2.1. Study Area

The study area is located in the YRD, northern Shandong Province, China, and consists of an alluvial plain formed at the estuary of the Yellow River, with Dongying City at its core (Figure 1). It encompasses a total area of approximately 5.45 × 10^5^ ha, including tidal flats, wetlands, and water bodies, with the land area covering about 2.3 × 10^5^ ha. The delta continues to expand into the Bohai Sea through sediment deposition at an annual rate of 20–30 × 10^2^ ha. The region is characterized by a warm–temperate, semi-humid, continental monsoon climate, with annual precipitation ranging from 550 to 650 mm, a mean temperature of 12.5–13.5 °C, and a frost-free period of ~210 days. The terrain of this area is predominantly low-lying (elevation: 2–10 m), featuring Yellow River alluvial plains, with soils classified as fluvo-aquic and saline, exhibiting localized salinization issues. Vegetation is primarily composed of herbaceous species, with Asteraceae, Fabaceae, and Poaceae as the dominant plant families. Representative species include *Suaeda salsa, Phragmites australis*, *Typha orientalis*, *Sonchus oleraceus*, *Imperata cylindrica*, *Limonium gmelinii*, etc. (Table A1).

### 2.2. Sampling Design and Vegetation Survey

To collect more herbaceous plant samples in wetlands of the YRD, five representative areas were selected for investigation and sampling from 21–30 September 2023 (Figure 1). The selection of sampling sites was primarily based on geographic information system (GIS) spatial analysis, covering major wetland types in the delta (natural vs. artificial, freshwater vs. saline, permanent vs. seasonal inundation, etc.); additionally, accessibility and sampling feasibility were taken into consideration. The specific sampling sites were selected as follows: The Yiqian’er Management Station of the Yellow River Delta National Nature Reserve (YRD-NNR) constitutes a river–swamp complex wetland (permanent freshwater wetland) featuring diverse wetland vegetation types (Figure 1A); the Huanghekou Management Station of the YRD-NNR represents an intertidal salt marsh wetland (permanent saline wetland) with typical coastal halophytic vegetation (Figure 1B); while the Dawenliu Management Station of the YRD-NNR is an estuarine freshwater marsh wetland (seasonally flooded wetland), dominated by reeds and other freshwater vegetation (Figure 1C). The Dongying Modern Agricultural Demonstration Zone serves as an agricultural constructed wetland (semi-permanent freshwater wetland) cultivating various saline–alkaline tolerant crops (Figure 1D). The Dongying Salt Botanical Garden is an artificial marsh wetland (permanent freshwater wetland, partially covered by saline–alkaline soil) created for the selection of salt-tolerant plants and to provide ornamental value, containing a considerable number of halophytes for sightseeing (Figure 1E). These areas boast abundant plant resources, including ferns, gymnosperms, and angiosperms. At each sampling area, two 300 m × 300 m plots were set up, with five 100 m × 100 m grass layer plots established at the center and four corners of each plot. The names of family, species, and genus of the annual and perennial herbaceous plants were photographed, recorded, and classified (Table 1).

### 2.3. Sample Collection and Determination

In each sampling plot, healthy herbaceous plant leaves and stems were systematically collected and categorized as either annual or perennial species, following the classification criteria in Flora of China. For each species, five healthy individuals were collected, resulting in a total of 17 annual herbaceous species (85 individuals, 170 leaf and stem samples) and 27 perennial herbaceous species (135 individuals, 270 leaf and stem samples). To minimize individual variations and ensure sample representativeness, samples of the same organ from the same species were pooled to form one composite sample for subsequent chemical analysis. All samples were immediately labeled after collection, stored in sealed plastic bags, and promptly transported to the laboratory. The plant samples (leaves and stems) were thoroughly rinsed with deionized water, deactivated in an oven at 105 °C for 30 min, and then continuously dried at 65 °C until a constant weight was achieved. The dried samples of different species and organ types were separately ground, sieved through an 80-mesh sieve, and sealed in self-sealing bags for storage pending nutrient content determination. Soil samples were collected from 5 randomly selected points in each plot. After removing surface litter, soil samples were taken from a depth of 0–10 cm using a soil auger with an inner diameter of 38 mm. The samples were transported to the laboratory for storage. After air-drying, soil samples were sieved through a 100-mesh screen after removing stones, fine roots, and other debris.

The C and N contents of both plant and soil samples were determined using an elemental analyzer (VARIO MACRO CUBE, Hesse, Germany). P content in plant and soil samples was digested and extracted using a mixture of concentrated sulfuric acid and perchloric acid, while available phosphorus (AP) in soil was extracted with NaHCO_3_ (pH 8.5). Both P and AP concentrations were measured using the Mo-Sb colorimetric method with an ultraviolet spectrophotometer (Persee, T700AS, Hangzhou, China). Soil pH was quantified with a pH meter (Mettler Toledo, FE28-Standard, Zurich, Switzerland) and soil electrical conductivity (EC) was measured with an EC meter (DDBJ-350F, Leici, Shanghai, China) in a 1:5 soil-to-water solution.

### 2.4. Data Analysis

All data are presented as mean ± standard error (SE). Elemental stoichiometric ratios were calculated as atomic ratios by converting units of C, N, and P weights in soil and plant to mol/kg or mmol/kg. To test the homogeneity of variance, all data were log10-transformed and assessed using Levene’s test. One-way analysis of variance (ANOVA) was performed to test for significant differences in C, N, and P contents and ecological stoichiometric ratios among annual and perennial plants, as well as their stem and leaf organs. Multiple comparisons were then made using Tukey’s post hoc test or the non-parametric Kruskal-Wallis test for heterogeneous variances.

Reduced major axis regression (RMA) was used to analyze the relationship between N and P for heterozygous growth, and the data were log10-transformed before analysis, which was calculated using the following equation(1)log10y=b·log10x+log10a
where y and x represent N and P contents, respectively; a is the allometric constant; *b* is the slope, as the allometric growth exponent, which indicates the rate of change between N and P. When *b* = 1, it signifies an isometric growth relationship between N and P contents; when *b* > 1 or *b* < 1, it signifies an allometric growth relationship.

A random forest model was applied to assess the relative contributions of nutrient allocation in plant organs to plant survival strategies. This statistical learning model generates accurate predictions and interpretations. Subsequently, redundancy analysis (RDA) and Pearson correlation analysis were conducted to explore the relationships between C, N, and P stoichiometric ratios, plant organs, and soil physicochemical properties between different life forms.

All statistical analyses were performed using SPSS 2017 (IBM Corp, Armonk, NY, USA) and R 4.1.2 2023 (R Core Team, Vienna, Austria), and all figures were drawn using Origin 2022 (Origin Lab, Northampton, MA, USA).

## 3. Results

### 3.1. Characterization of C, N, and P Stoichiometry in Herbs of Different Life Forms

Figure 2 shows that the C, N, and P contents of herbaceous plants in the study area were 408.4 g/kg, 20.2 g/kg, and 2.3 g/kg, respectively, with significant differences existing in C, N, and P contents and their stoichiometric ratios between annual and perennial herbaceous plants. Specifically, the C, N, and P contents in annual herbaceous plants were 331.5 g/kg, 26.0 g/kg, and 2.7 g/kg, respectively, while in perennial herbaceous plants, the corresponding values were 439.0 g/kg, 18.5 g/kg, and 2.1 g/kg. Annual herbaceous plants have higher N and P contents compared to perennial herbaceous plants, while their C content is significantly lower (*p* < 0.05).

Additionally, there are significant differences in the stoichiometric ratios of C, N, and P (including R_CN_ and R_CP_) between the two groups. For annual herbaceous plants, the mean values of R_CN_, R_CP_, and R_NP_ were 18.8, 339.6, and 22.0, respectively, while for perennial herbaceous plants, these values were 42.2, 595.8, and 19.9, respectively. It is evident that perennial herbaceous plants, with a lower R_NP_, exhibit higher R_CN_ and R_CP_ compared to annual herbaceous plants. However, there is no statistically significant difference in R_NP_ between them. The random forest analysis indicates (Figure 3) that the C, N, and P contents of plant organs and their stoichiometric ratios have different impacts on annual and perennial herbaceous plants. Annual herbaceous plants are more sensitive to the C content in stems, the C content in leaves, and the R_CP_ in stems, whereas perennial species are only sensitive to the C content in stems.

### 3.2. Characterization of C, N, and P Content and Distribution in Stem and Leaf Organs

Significant differences in the distribution of nutrient elements between the stem and leaf organs were observed in the YRD, as shown in Figure 4. The C and N contents in the leaves of herbaceous plants were significantly higher than those in the stems (*p* < 0.05). Though the P content in the leaves was higher than in the stems, the difference was not statistically significant. The R_CN_ and R_CP_ values of the leaves were 20.8 and 486.9, respectively, both lower than those of the stems, whereas the R_NP_ value of the leaves was higher than that of the stems.

Significant differences were observed in the log-linear correlation of N and P contents between leaf and stem organs (Table 2). Specifically, the leaf N vs. P scaling exponent was 0.62 (*r*^2^ = 0.16, *p* < 0.05), while the log-linear correlation of N and P contents in the stems was weak and not significant, with a low explanatory rate for the stem N vs. P scaling exponent. Moreover, differences in the N and P contents ratio exponents were also observed between annual and perennial plants. The log-linear correlation between N and P contents was not significant in whole-plant analyses for both annual and perennial herbaceous species. However, the log-linear correlation of N and P contents in the leaves of perennial herbaceous plants was significant, with a scaling exponent of 0.56 (*r*^2^ = 0.14, *p* < 0.05). Therefore, the log-linear correlation of N and P contents was strongly associated with the organ type, which may highlight that leaves, as the metabolic centers of plants, were subject to stronger physiological constraints in terms of elemental coupling compared to other organs (such as stems).

### 3.3. Correlation of Soil Physicochemical Factors with Plant C, N, and P Stoichiometric Characteristics

Soil physicochemical factors exerted significant effects on the C, N, and P contents and their stoichiometric ratios in both annual and perennial herbaceous plants (Table 3). RDA further revealed that soil physicochemical factors collectively explained 78.43% of the variation in stoichiometric traits of plant leaf and stem C, N, and P in the YRD region (Figure 5), with soil P content, soil N content, and soil pH identified as the main driving factors influencing herbaceous plant stoichiometry.

In areas dominated by annual herbs, soil AP content was negatively correlated with leaf P content but positively correlated with the R_CP_ in both leaves and stems. Soil C content showed a positive correlation with the leaf R_CN_. Meanwhile, soil EC was negatively correlated with stem C content, whereas soil pH was positively correlated with stem C content. In perennial plant-dominated areas, soil N content was negatively correlated with leaf P content but positively correlated with leaf R_CP_. A significant positive correlation was observed between stem R_CN_ and soil EC. These results collectively indicate that soil properties are important factors influencing the stoichiometric characteristics of herbaceous plants in the study area.

## 4. Discussion

### 4.1. Nutrient Characteristics and Adaptive Strategies of Herbaceous Plants of Different Life Forms in the YRD

In the YRD, herbaceous plants exhibited lower C content compared not only to those in Poyang Lake wetlands and the Ebinur Lake Basin in Xinjiang, but also to the global average for herbaceous plants, while their N and P contents were higher than the average levels in wetland plants in China [27,28,29,30]. This trait reflects that plants in the YRD region have weaker C sequestration capacity but stronger N uptake ability, indicating the development of distinct adaptive survival strategies under environmental pressures compared to plants in other regions. High salinity suppresses photosynthesis and consequently reduces C assimilation capacity, while tidal and flooding stresses further stimulate plants to synthesize more protective enzymes and proteins, thereby increasing their demand for N and P [6,31]. The study also found that perennial herbs had higher C content but lower N and P contents compared to annual herbs, which is consistent with findings on riparian plants in the Three Gorges Reservoir [32]. In terms of stoichiometric ratios, annual herbs exhibited lower R_CN_ and R_CP_ but higher R_NP_ than perennial herbs. These findings suggested that perennial herbs possess a stronger capacity for C assimilation, while annual herbs display faster growth rates and nutrient uptake efficiency, reflecting their distinct survival strategies [15,33,34].

Annual plants have short life cycles, and their survival depends on rapidly completing vegetative growth and reproduction within a single growing season [32,35]. Consequently, they allocate absorbed resources preferentially to highly metabolically active photosynthetic tissues and reproductive organs, which are rich in nucleic acids and proteins [36]. This leads to elevated overall N and P concentrations, resulting in lower R_CN_ and R_CP_. This strategy reflects their approach of rapid nutrient uptake and efficient utilization to adapt to frequently disturbed habitats. In contrast, the core strategy of perennial plants is long-term survival and stress resistance. To endure multi-year stresses such as saline-alkali conditions and waterlogging, they prioritize resource allocation toward building persistent structures and synthesizing defensive compounds [37,38]. These structural and defensive tissues are characterized by high C content and very low N and P contents. Additionally, perennial plants store resources in underground organs for sprouting in the following year, and these storage substances are primarily C-based compounds [35]. This significant investment in high C, low N, and low tissue density dilutes the overall N and P concentrations in their bodies, leading to higher R_CN_ and R_CP_. Although this strategy sacrifices short-term growth rates, it enhances long-term tolerance to nutrient-poor and stressful environments. From a chemical ecology perspective, this explains the dominant distribution of perennial plants in the study area.

### 4.2. Organ-Specific Nutrient Allocation Strategies and Their Role in Environmental Adaptation

Plant nutrient allocation strategies across different organs reflect mechanisms of environmental adaptation [20,39]. In the YRD, we observed a significantly higher N content in leaves compared to stems (*p* < 0.05), along with marginally greater C and P contents in the leaves, which contrasts with the typical nutrient allocation patterns observed in terrestrial plants nationwide [5]. This is primarily because, under salt stress conditions, plants adopt a resource allocation strategy that prioritizes critical functions. As the core organ for photosynthesis and metabolic regulation, leaves require the accumulation of substantial N amounts to synthesize chlorophyll, enzymatic proteins, and osmoregulatory substances, thereby maintaining physiological activity in high-salinity environments [34,40]. In contrast, stems primarily serve mechanical support functions and exhibit lower metabolic activity. Consequently, when nutrients are limited, plants preferentially reduce N allocation to stems, resulting in lower N content in these organs. Simultaneously, to cope with salt stress, plants accumulate non-structural carbohydrates such as starch and soluble sugars in leaves to enhance osmoregulatory capacity, while stems reduce structural carbon investment to minimize overall resource consumption. Ultimately, this leads to the formation of a coordinated adaptive strategy characterized by “high osmotic protection in leaves and low structural maintenance in stems”.

Previous studies have demonstrated that variations in plant C, N, and P stoichiometry among different organs can directly reflect their functional differentiation and mechanisms of environmental adaptation [41]. The significant differences in nutrient allocation strategies at the organ level are driven by the distinct functions of leaves and stems [42]. The R_NP_ of leaves and stems in the study area was significantly higher than the national average for terrestrial plants, indicating that herbaceous plants in the YRD are primarily limited by P. RMA analysis of leaf N and P contents (Table 2) revealed a scaling exponent of 0.62, suggesting that under P-limited conditions, plants preferentially allocate P to leaves. Prioritizing P allocation to leaves ensures sufficient energy metabolism in these vital organs [43]. The lower R_NP_ in stems, primarily structural organs, further demonstrates how plants achieve adaptive regulation through organ-specific resource allocation strategies [44]. This differential allocation between leaves and stems represents an efficient physiological trade-off that optimizes whole plant performance in P-limited environments. This synergistic adaptation pattern among organs directly shapes the plant’s overall life history strategy. Random forest analysis (Figure 3) revealed that the key factor influencing the life history strategies of annual versus perennial plants was C content, particularly in stems, which aligns with the whole-plant economic spectrum theory proposed by Freschet et al. [45,46]. The whole-plant economic profiling theory proposes that plants have evolved a holistic resource allocation strategy through multi-organ coevolution [47,48]. Our results indicate that stem C content is a key pivot for balancing the “fast-conservative” trade-off. RDA further revealed a negative correlation between stem C content and leaf N content, elucidating the mechanism of resource allocation trade-offs among organs (Figure 5). Perennials adopt a conservative strategy of “high stalk C + low leaf N” to enhance stress tolerance, whereas annuals adopt a “low stalk C + high leaf N” rapid strategy to maximize photosynthetic efficiency. These strategic differences not only validate the core tenets of the whole-plant economics spectrum theory but also uncover the critical regulatory role of inter-organ nutrient allocation in plant environmental adaptation.

### 4.3. Soil Physicochemical Properties and Their Impact on C, N, and P Allocation in Plant Organs

Plants often adjust their stoichiometric characteristics in response to changes in soil conditions [34,49]. In our study, soil physicochemical factors collectively explained 78.43% of the variation in C, N, and P stoichiometric characteristics of herbaceous plant stems and leaves, indicating that these soil physicochemical properties play a crucial role in plant growth and ecological adaptation. It is noteworthy that the YRD is undergoing intense environmental changes, such as increasing salinization and frequent human activities. These changes may indirectly affect the stoichiometric relationships in plant–soil systems by altering soil salinity, nutrient availability, and water conditions, which could partially explain the variations in soil-plant stoichiometric relationships observed in this study. Aligning with previous studies, we also found a positive correlation between soil P and P concentrations in both stems and leaves [7,50]. Meanwhile, the correlation between soil N and plant N concentration was weaker [44,50]. These results suggested that plant P is more strongly influenced by soil availability, whereas plant N is primarily regulated by functional group characteristics.

Furthermore, soil pH had significant effects on plant organ stoichiometry in our study. As a comprehensive indicator of salinity and alkalinity, soil pH significantly influences various soil properties, including ion content, nutrient transformations, and microbial activity, all of which affect soil fertility, organic matter decomposition, N mineralization, and ultimately, plant growth [51,52,53]. Previous studies have shown that when soil pH falls below 6, the activity of N-fixing bacteria decreases, while nitrification is inhibited when pH exceeds 8 [54,55]. Optimal N availability occurs within the pH range of 6–8, with P availability peaking between pH 6.5–7.5. Although the soil pH (7.5–8.5) in the study area theoretically falls within the optimal range for N and P availability, it showed a significant correlation only with stem C in annual plants and no significant correlation with N and P contents in plants overall. This may be attributed to inherent chemical constraints within the soil despite the favorable pH range. In mildly alkaline soil environments, soluble P readily reacts with calcium ions to form insoluble calcium phosphates, leading to P immobilization [56]. For N, ammonia volatilization losses from ammonium-based fertilizers and leaching losses of nitrate are particularly pronounced under alkaline conditions [57]. Concurrently, the reduced availability of trace elements such as iron and zinc may constrain N assimilation efficiency in plants. On the other hand, annual plants, with their shallow root systems, are more susceptible to fluctuations in topsoil pH. This may drive them to adjust structural C synthesis in stems as a stress response. In contrast, perennial plants, leveraging their deep root systems and symbiotic microbial associations, can mitigate the impacts of pH variations, thereby exhibiting lower sensitivity [58].

The positive correlation between R_NP_ in stems and leaves and soil pH indicates that plant P is more sensitive to soil salinity stress. Under nutrient limitation, P content in plant stems and leaves shows heightened sensitivity to variations in soil P. This sensitivity reflects the adaptive strategies of plants to optimize P acquisition and allocation in P-deficient soils [59]. Our previous research showed the YRD ecosystem experiences N limitation, with P becoming the secondary limiting factor in N-limited systems [60,61]. In response to N deficiency, plants adjust their resource allocation strategies. Similarly, Güsewell revealed that in N-limited systems, plants increase their demand for P to maintain metabolic balance [33]. This occurs because N deficiency renders P a critical limiting factor for plant growth. When soil P availability increases, plants enhance C metabolism and elevate the content of soluble sugars and phosphates to improve salinity stress adaptation. Additionally, integrating correlation analyses and RDA results (Table 3 and Figure 5) revealed that soil physicochemical factors had stronger effects on plant organ stoichiometry than plant life forms. This suggests these edaphic factors indirectly influence plant life history strategies by regulating organ-level stoichiometric characteristics.

## 5. Conclusions

This study investigated the C, N, and P stoichiometric characteristics of herbaceous plants in the YRD, focusing on annual vs. perennial life forms and their organ-specific nutrient allocation. Results revealed that plants in the YRD have unique survival strategies to thrive in saline-alkaline conditions. Leaf tissues had higher N and P contents than stems, indicating that plants prioritize allocating nutrient resources to leaves to cope with salt stress. Annual plants exhibited a low-C, high-R_NP_ strategy with lower R_CP_, facilitating rapid growth but increasing sensitivity to environmental stress, while perennial plants adopted a high-C, low-R_NP_ strategy with elevated R_CP_ to enhance stress resistance. Soil properties played a significant role in shaping plant organ stoichiometry, and the correlation between them was stronger than compared to the correlation between soil physicochemical properties and plant life forms. These findings suggest that in saline-alkaline environments, plants rely on soil factors to indirectly modulate their life history strategies through organ-level nutrient allocation. This highlights the importance of organ-specific nutrient partitioning for coastal wetland restoration, emphasizing the need to optimize plant selection based on nutrient allocation patterns. The study will contribute to understanding plant adaptations and support sustainable conservation efforts in the coastal wetlands of the YRD.

## Figures and Tables

**Figure 1 biology-14-01132-f001:**
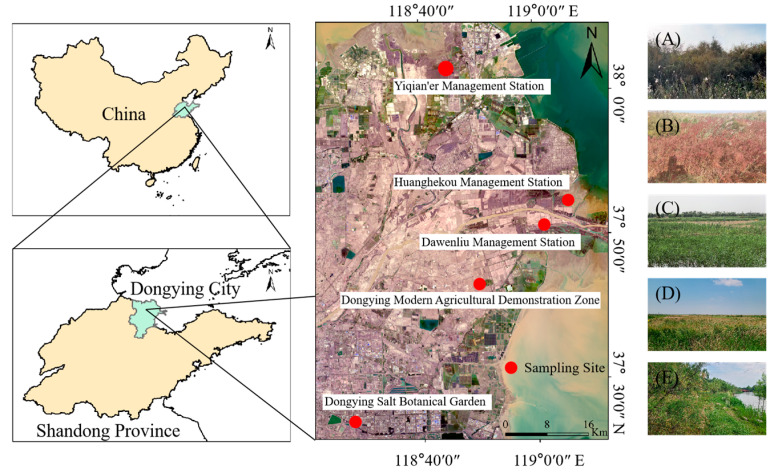
Ketch map of sampling sites in the Yellow River Delta region, China. (**A**) Field photograph of sampling site at Yiqian’er Management Station; (**B**) Field photograph of sampling site at Huanghekou Management Station; (**C**) Field photograph of sampling site at Dawenliu Management Station; (**D**) Field photograph of sampling site at Dongying Modern Agricultural Demonstration Zone; (**E**) Field photograph of sampling site at Dongying Salt Botanical Garden.

**Figure 2 biology-14-01132-f002:**
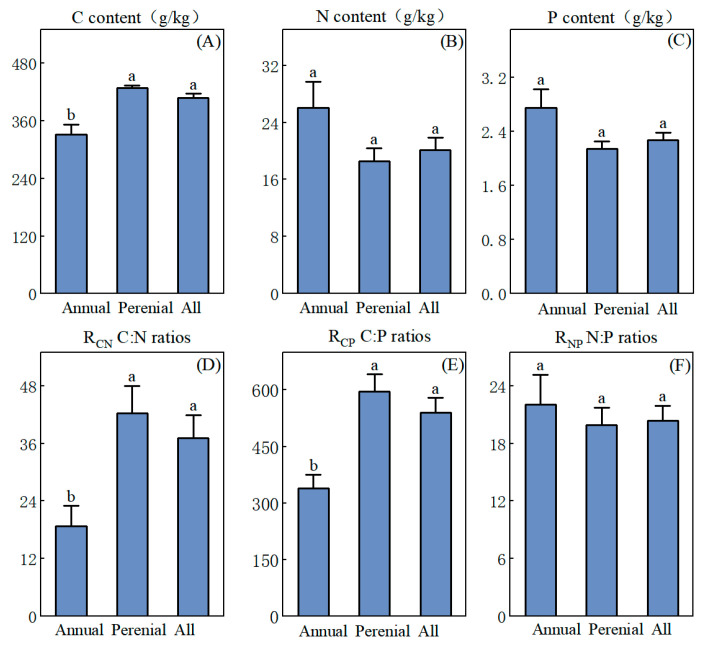
The contents of C, N, and P (**A**–**C**) and their stoichiometric ratios (**D**–**F**) in different life forms of plants in the YRD, China. Error bars represent the standard error (SE) of the mean. Different lowercase letters denote significant differences among life forms.

**Figure 3 biology-14-01132-f003:**
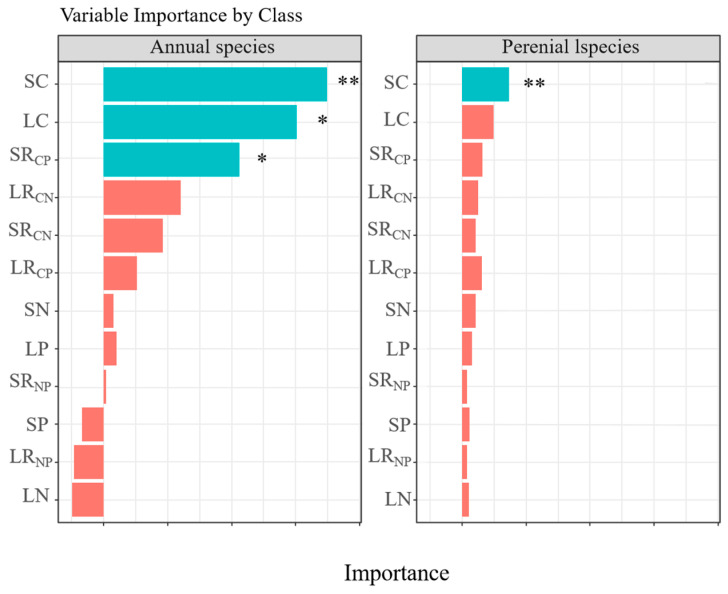
Random forest analysis of C, N, and P stoichiometric characteristics in different life forms and organs of plants. Note: The results are color-coded, with the blue bars representing significant differences and the red bars indicating a lack of statistical significance. In the box plot, LC, leaf C content; LN, leaf N content; LP, leaf P content; LR_CN_, leaf C:N ratios; LR_CP_, leaf C:P ratios; LR_NP_, leaf N:P ratios; SC, stem C content; SN, stem N content; SP, stem P content; SR_CN_, stem C:N ratios; SR_CP_, stem C:P ratios; SR_NP_, stem N:P ratios; * and ** indicate that the feature significantly impacts model prediction.

**Figure 4 biology-14-01132-f004:**
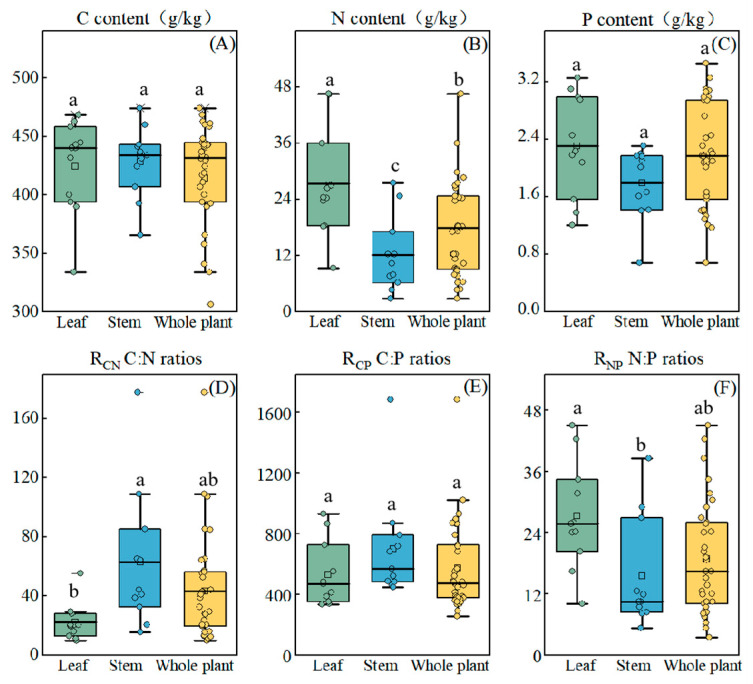
The contents of C, N, and P (**A**–**C**) and their stoichiometric ratios in leaf, stem, and whole plant (**D**–**F**) in the Yellow River Delta in China. Different lowercase letters denote significant differences among life forms.

**Figure 5 biology-14-01132-f005:**
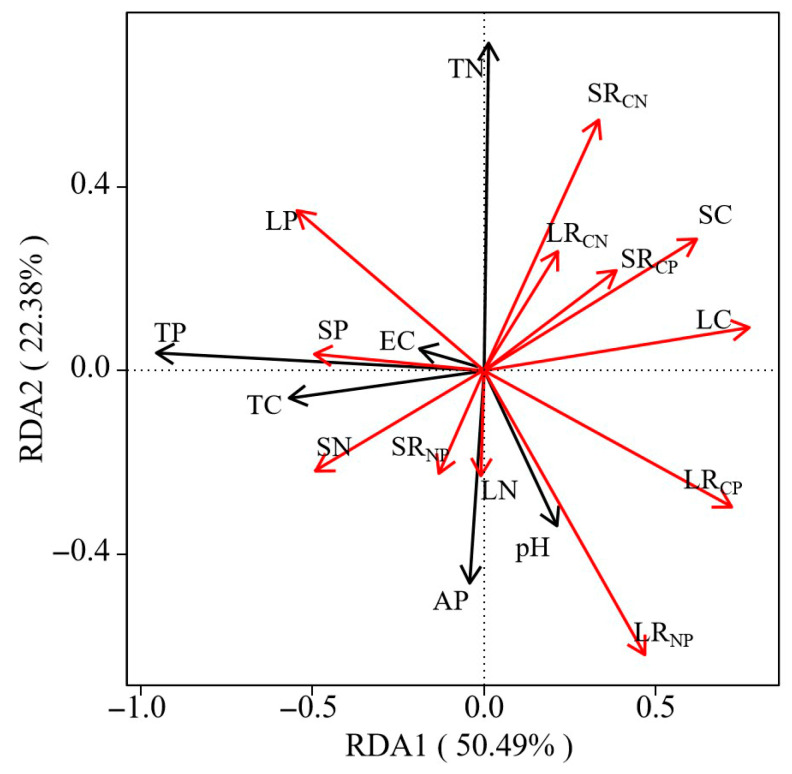
Redundancy analysis (RDA) between soil ecological stoichiometric characteristics and nutrients in plant stem and leaf organs. Note: The black arrow shows soil ecological stoichiometric characteristics, and the red arrow shows nutrient concentration in various organs. In the box plot, LC, leaf C content; LN, leaf N content; LP, leaf P content; LR_CN_, leaf C:N ratios; LR_CP_, leaf C:P ratios; LR_NP_, leaf N:P ratios; SC, stem C content; SN, stem N content; SP, stem P content; SR_CN_, stem C:N ratios; SR_CP_, stem C:P ratios; SR_NP_, stem N:P ratios; TC, soil total C content; TN, soil total N content; TP, soil total P content; AP, soil available phosphorus content; EC, soil electrical conductivity; pH, soil pH value.

**Table 1 biology-14-01132-t001:** Plant list.

Life Form	Family	Species
Annual herbaceous plant	*Asteraceae*	*Conyza canadensis*
	*Asteraceae*	*Sonchubrachyotus*
	*Asteraceae*	*Artemisia annua*
	*Asteraceae*	*Anthemis cotula*
	*Poaceae*	*Eragrostis minor*
	*Poaceae*	*Digitaria sanguinali*
	*Poaceae*	*Echinochloa crus-galli*
	*Poaceae*	*Eleusine indica*
	*Poaceae*	*Setaria viridis*
	*Amaranthaceae*	*Suaeda salsa*
	*Amaranthaceae*	*Salicornia europaea*
	*Cannabaceae*	*Humulus scandens*
	*Linaceae*	*Linum usitatissimum*
	*Portulacaceae*	*Portulaca oleracea*
	*Cucurbitaceae*	*Cucumis melo* var. *agrestis*
	*Euphorbiaceae*	*Euphorbia hypericifolia*
	*Fabaceae*	*Sesbania cannabina*
Perennial herbaceous plant	*Poaceae*	*Imperata cylindrica*
	*Poaceae*	*Phragmites australis*
	*Poaceae*	*Deyeuxia pyramidalis*
	*Poaceae*	*Cynodon dactylon*
	*Poaceae*	*Phalaris arundinacea*
	*Fabaceae*	*Sphaerophysa salsula*
	*Fabaceae*	*Medicago sativa*
	*Fabaceae*	*Medicago lupulina*
	*Asteraceae*	*Artemisia mongolica*
	*Asteraceae*	*Artemisia scoparia*
	*Asteraceae*	*Lactuca tatarica*
	*Asteraceae*	*Sonchus arvensis*
	*Asteraceae*	*Achillea millefolium*
	*Asteraceae*	*Cirsium setosum*
	*Plantaginaceae*	*Plantago major*
	*Apocynaceae*	*Apocynum venetum*
	*Malvaceae*	*Alcea rosea*
	*Iridaceae*	*Iris lactea* var. *chinensis*
	*Iridaceae*	*Iris tectorum*
	*Asphodelaceae*	*Hemerocallis fulva*
	*Lamiaceae*	*Scutellaria baicalensis*
	*Plumbaginaceae*	*Limonium gmelinii*
	*Oleaceae*	*Syringa oblata Lindl*
	*Typhaceae*	*Typha orientalis*
	*Cyperaceae*	*Bolboschoenus maritimus*
	*Asclepiadaceae*	*Cynanchum chinense*
	*Asparagaceae*	*Asparagus officinalis*
	*Amaranthaceae*	*Kochia scoparia* var. *sieversiana*
	*Caprifoliaceae*	*Lonicera japonica*
	*Lamiaceae*	*Salvia miltiorrhiza*

**Table 2 biology-14-01132-t002:** Summary of RMA N versus P regression results of log10-transformed data across herbaceous plants and organs of different life forms.

Plant	Organ	Intercept	Exponent	95%CI	*r* ^2^	*p*
All Species	Leaf	1.174	0.618	(0.03–1.28)	0.169	0.02
	Stem	0.96	0.35	(−1.12–2.1)	0.014	0.3
Annual Species	Leaf	1.242	0.541	(−2.23–5.57)	0.16	0.28
	Stem	1.067	0.524	—	0.006	0.44
Perennial Species	Leaf	1.183	0.566	(−0.18–1.38)	0.14	0.04
	Stem	1.517	1.388	(−1.51–1.19)	0.003	0.47

**Table 3 biology-14-01132-t003:** Pearson correlation coefficients between the main biogenic elements of the different life forms of herbaceous plants and common physicochemical characteristics of soil.

Index		Soil					
		pH	EC	TC	TN	TP	AP
Annual plants	LC	−0.265	0.277	−0.023	−0.223	−0.75	0.549
	LN	−0.572	0.519	−0.781	−0.879 *	−0.284	0.072
	LP	−0.662	0.73	−0.164	−0.365	0.582	−0.926 *
	L_CN_	0.686	−0.613	0.888 *	0.944 *	−0.104	0.443
	L_CP_	0.642	−0.676	0.343	0.483	−0.598	0.910 *
	L_NP_	0.007	−0.142	−0.682	−0.555	−0.614	0.646
	SC	0.919 *	−0.950 *	0.317	0.661	0.217	0.404
	SN	−0.333	0.357	−0.542	−0.526	0.696	−0.569
	SP	0.289	−0.263	0.286	0.427	0.738	−0.567
	S_CN_	0.764	−0.746	0.712	0.848	−0.248	0.601
	S_CP_	0.652	−0.713	−0.057	0.178	−0.454	0.976 **
	S_NP_	−0.409	0.428	−0.647	−0.695	0.282	−0.166
Perennial plants	LC	−0.024	0.162	0.104	0.377	−0.09	0.298
	LN	0.096	−0.319	−0.276	−0.189	−0.159	−0.105
	LP	−0.154	−0.095	−0.402	−0.532 *	0.003	−0.24
	L_CN_	−0.086	0.31	0.091	0.041	−0.058	0.031
	L_CP_	0.108	0.043	0.276	0.565 *	−0.106	0.278
	L_NP_	0.201	−0.311	0.023	0.288	−0.125	0.092
	SC	0.163	0.117	0.222	0.369	−0.12	0.341
	SN	−0.178	−0.17	−0.253	−0.293	−0.139	−0.088
	SP	−0.099	−0.164	−0.053	−0.31	0.447	−0.387
	S_CN_	−0.083	0.566 *	0.132	0.269	−0.144	0.116
	S_CP_	−0.03	0.466	0.253	0.41	−0.318	0.418
	S_NP_	−0.002	−0.117	−0.17	−0.129	−0.293	0.053

* *p* < 0.05, ** *p* < 0.01. LC, leaf C content; LN, leaf N content; LP, leaf P content; LR_CN_, leaf C:N ratios; LR_CP_, leaf C:P ratios; LR_NP_, leaf N:P ratios; SC, stem C content; SN, stem N content; SP, stem P content; SR_CN_, stem C:N ratios; SR_CP_, stem C:P ratios; SR_NP_, stem N:P ratios; pH, soil pH value; EC, soil electrical conductivity; TC, soil total C content; TN, soil total N content; TP, soil total P content; AP, soil available phosphorus content.

## Data Availability

The raw data supporting the conclusions of this article will be made available by the corresponding authors on request.

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
