# Peer review of "Ecological Stoichiometric Characteristics and Adaptive Strategies of Herbaceous Plants in the Yellow River Delta Wetland, China"

_biology, 2025, doi:10.3390/biology14091132_

Round 1
Reviewer 1 Report
Comments and Suggestions for Authors
In this study, the authors study the ecological stoichiometric characteristics of the herbaceous plants in YRD wetland. This is a typical stoichiometric study with traditional methods and unsurprising results, but is worthy of publication. Discussing the environmental adaptation mechanisms of plants based on elemental stoichiometry will never become outdated. In general, the methodology is accurate, the logical structure is reasonable, the language is fluent in this manuscript. Unfortunately, the study only includes the three most commonly used elements, C, N and P. Furthermore, my major concerns are the explanation of the results, especially the effects of soil parameters on plant elements. The other one is the effects of phylogeny (species differences) on the plant element concentration. Detailed comments are as follows.
Title
Too neutral. The title should capture the key findings of the article.
Introduction
L79-81, It would be much better if authors can give clear hypotheses in the introduction section.
Methods
Legend of Figure 1 is too verbose.
L109, Sampling date is at the end of September, the end of the growing season, if so, the element concentration will be affected, especially annual plants.
L124, 300 ´ 300 m2 or 100 ´ 100 m2 is incorrect, it should be 300m ´ 300m or 100m ´ 100m, if I haven’t misunderstood.
Table 1, the column of “Genus” is needless.
L131-133, the description of the samples for each species is not clear. How many individuals were collected for each species? five? Need a clearer expression.
L143, “Each sample was labeled and numbered accordingly.” Similar descriptions are unnecessary. There are still some other sentences in the article, e.g. L135.
Results
L211-212, I can understand why leaf N>stem N, but how to explain the C? The C concentration of stem (mechanical tissue) should be higher than that of leaves.
Figure 4, how to determine the element concentration of the whole plant? Measure the element concentrations of the leaves and stems separately, calculate the average value? Or measure the samples after mixing the leaves and stems together? In addition, mark the symbols indicating significant differences in Figure 4.
Table 2, calculated the parameters (exponent, intercept) of N-P regressions, but did not explain the differences of the parameters between different life forms and organs.
L233, when discuss the effects of soil properties on plant elements, it should be “regression” rather than “correlation”.
Some results cannot be understood easily. For example, soil AP negatively correlated with leaf P? and the results from L239 to L243.
From Table 3, there are not many soil properties that have significant effects on the plant element concentrations.
Discussion
L269-270, the data at global and national scales (Reich and Oleksyn 2004, Tang et al. 2018) are mainly consist of woody plant, and cannot be directly compared.
L284-287, the results in this study did not include the calculation of CSR strategy, and just relying on element concentration alone cannot determine the CSR strategy. So, discussing the CSR here is not appropriate.
Author Response
Comments 1: In this study, the authors study the ecological stoichiometric characteristics of the herbaceous plants in YRD wetland. This is a typical stoichiometric study with traditional methods and unsurprising results, but is worthy of publication. Discussing the environmental adaptation mechanisms of plants based on elemental stoichiometry will never become outdated. In general, the methodology is accurate, the logical structure is reasonable, the language is fluent in this manuscript. Unfortunately, the study only includes the three most commonly used elements, C, N and P. Furthermore, my major concerns are the explanation of the results, especially the effects of soil parameters on plant elements. The other one is the effects of phylogeny (species differences) on the plant element concentration. Detailed comments are as follows.
Response 1: We would like to thank the reviewer for their thoughtful and constructive feedback. Below, we have addressed the specific comments (from comments 2 to 17) and suggestions. We hope that these revisions address the concerns raised and improve the overall quality of the manuscript. Once again, we would like to express our gratitude for the reviewer’s insightful feedback. We believe the manuscript is now stronger and more aligned with the expectations for publication.
Comments 2: Title
Too neutral. The title should capture the key findings of the article.
Response 2: Thank you very much for taking the time and effort to review our manuscript and providing highly constructive feedback. We fully agree with your perspective regarding the title-one that directly highlights the key findings of the study would undoubtedly enhance the impact of the article. In response, we have revised the title to more explicitly reflect the key findings of our study: "Soil-Driven Nutrient Allocation and Saline-Alkali Adaptation of Herbaceous Plants in the Yellow River Delta Wetland, China." We believe this revised title better encapsulates the core aspects of our research. Given that two other reviewers did not provide title modification suggestions, we will presented this revised title for the editor’s consideration.
Comments 3: Introduction
L79-81, It would be much better if authors can give clear hypotheses in the introduction section.
Response 3: We sincerely thank the reviewer for this constructive suggestion. We fully agree that stating clear hypotheses will strengthen the focus of our introduction. Accordingly, we have added relevant content at the end of the introduction section (please see lines 93-97, Page 3) to explicitly state our research hypotheses.
Comments 4: Methods
Legend of Figure 1 is too verbose.
Response 4: Thank you for the reviewer's suggestion. We have simplified the caption of Figure 1 accordingly.
Comments 5: Methods
L109, Sampling date is at the end of September, the end of the growing season, if so, the element concentration will be affected, especially annual plants.
Response 5: Thank you for your important comment. We fully recognize the potential influence of sampling timing (late growing season) on plant elemental concentrations. The decision to sample in late September was based on the following research considerations: this period coincides with plants completing key growth stages, wherein their elemental concentrations best reflect the ultimate resource allocation strategies of the entire growing season-annual plants are occupied with nutrient translocation for reproduction, while perennial plants focus more on overwintering storage and stress-resistant structural development. This phenological stage precisely serves to highlight the differences in stoichiometric characteristics between the two life forms, aligning particularly well with our objective of studying plant adaptation strategies.
Comment 6: Methods
L124, 300 × 300 m² or 100 × 100 m² is incorrect, it should be 300 m × 300 m or 100 m × 100 m, if I haven’t misunderstood.
Response 6: We sincerely apologize for the oversight. The notation has been corrected throughout the manuscript to “300 m × 300 m” and “100 m × 100 m” as suggested.
Comment 7: Methods
Table 1, the column of “Genus” is needless.
Response 7: We agree that the “Genus” column was redundant. It has been removed from Table 1 to improve the table’s clarity and focus on essential information.
Comment 8: Methods
L131-133, the description of the samples for each species is not clear. How many individuals were collected for each species? five? Need a clearer expression.
Response 8: Thank you for highlighting this lack of clarity. We have revised the sentence to explicitly state: “For each species, five healthy individuals were collected” (Line 157).
Comment 9: Methods
L143, “Each sample was labeled and numbered accordingly.” Similar descriptions are unnecessary. There are still some other sentences in the article, e.g., L135.
Response 9: We thank the reviewer for this suggestion. The redundant sentences, including the one in Line 143 and similar instances (e.g., Line 135), have been removed to make the methodology description more concise and focused. This change can be found on page 5, and lines 162-172.
Comments 10: Results
L211-212, I can understand why leaf N>stem N, but how to explain the C? The C concentration of stem (mechanical tissue) should be higher than that of leaves.
Response 10: Thank you very much for raising this insightful and valuable point, which has helped us further refine our understanding and interpretation of the results. Your observation that "stems, as mechanical tissues, generally exhibit higher C concentrations" is indeed a widely accepted understanding. In our initial analysis, we also carefully scrutinized and validated this result. We confirmed that the measurements are accurate and reproducible. We have included relevant explanations for this finding in the Discussion section.“Consequently, when nutrients are limited, plants preferentially reduce N allocation to stems, resulting in lower N content in these organs. Simultaneously, to cope with salt stress, plants accumulate non-structural carbohydrates such as starch and soluble sugars in leaves to enhance osmoregulatory capacity, while stems reduce structural C investment to minimize overall resource consumption.”
Comments 11: Results
Figure 4: how to determine the element concentration of the whole plant? Measure the element concentrations of the leaves and stems separately, calculate the average value? Or measure the samples after mixing the leaves and stems together? In addition, mark the symbols indicating significant differences in Figure 4.
Response 11: Thank you very much for raising this important question. In our study, the determination of whole-plant element concentrations was carried out using the "measure the samples after mixing the leaves and stems together". At the same time, we greatly appreciate your feedback regarding the chart annotations. We fully agree with your suggestion that the figures should clearly include symbols denoting statistical significance. This was an oversight during our manuscript preparation. We will promptly supplement all relevant charts in the revised manuscript with significance indicators.
Comments 12: Results
Table 2, calculated the parameters (exponent, intercept) of N-P regressions, but did not explain the differences of the parameters between different life forms and organs.
Response 12: Thank you very much for raising this valuable point. This comment is highly important and has prompted us to think more deeply about the differences in N-P relationships among different organs and plant life forms.
As you rightly noted, our results show that a significant log-linear N-P relationship exists only at the leaf level, whereas this relationship is weaker or non-significant in stems and at the whole-plant level. We hypothesize that this may be because leaves are the core organs for photosynthesis and metabolic activities in plants. As key components of photosynthetic proteins and energy carriers (such as ATP), N and P exhibit tight biochemical coupling in terms of demand and supply, resulting in a coordinated pattern of variation. In contrast, stems primarily serve supportive and transport functions, and their elemental composition is less constrained by metabolic coupling, leading to relatively independent variations in N and P concentrations.
Regarding the differences between life forms, annual and perennial plants may adopt distinct resource allocation strategies. The stable N-P relationship in perennial leaves may reflect their long-term maintenance of internal physiological homeostasis (a "conservative" strategy), while the flexible growth strategy of annual plants might make their elemental concentrations more sensitive to environmental feedback, thereby weakening the statistical correlation.
Although due to space limitations, we were unable to elaborate on this in the discussion section of the original manuscript, we have followed your suggestion and added the following sentence in line 274-276 of the paper to explain the significance of this finding: "Therefore, the log-linear correlation of N and P contents was strongly associated with the organ type, which may highlight that leaves, as the metabolic centers of plants, are subject to stronger physiological constraints in terms of elemental coupling compared to other organs (such as stems)."
Comments 13: Results
L233, when discuss the effects of soil properties on plant elements, it should be “regression” rather than “correlation”.
Response 13: Thank you for your valuable suggestion. We fully agree with your perspective that when exploring the influence of environmental factors on plant elements, methods such as regression analysis that can reveal dependencies between variables are far more interpretable than simple correlation analysis. In fact, in our preliminary analysis, we did perform correlation analysis to explore all pairwise relationships between variables. To address your point more deeply and comprehensively, we have followed your recommendation and adopted RDA-based on a multivariate regression framework as the core analytical method. RDA is particularly well-suited for our study, as it allows for a more integrated assessment of the overall influence of a set of environmental variables (such as various soil properties) on the entire plant element composition, while also quantifying the independent contribution of each environmental factor. This approach is more efficient and powerful than conducting multiple separate regression analyses. In accordance with your feedback, we have made the following modifications to the manuscript: In the Results section, we have repositioned the RDA results as the overarching conclusion to emphasize the combined explanatory power of multiple factors, while placing the specific correlation results later in the section to clarify their supporting role in relation to the main model findings, thereby improving logical flow. This change can be found on page 9, and lines 281-286.
Comments 14: Results
Some results cannot be understood easily. For example, soil AP negatively correlated with leaf P? and the results from L239 to L243.
Response 14: Thank you very much for your valuable feedback. You mentioned that it is difficult to interpret the results related to soil available phosphorus (AP) and plant P content as well as stoichiometric ratios, and we would like to provide the following explanation: In the study area, the soil AP content shows a negative correlation with leaf P content but a positive correlation with the RCP in leaves and stems. This seemingly paradoxical phenomenon reveals the unique P cycling mechanism in the saline-alkaline ecosystem of the Yellow River Delta, where a disconnect between the "chemical availability" and "biological availability" of soil P occurs in this specific habitat, triggering active physio-ecological adaptation strategies in plants. In the alkaline soil environment, the AP measured by chemical extraction methods readily combines with calcium ions to form insoluble calcium phosphate salts, leading to a separation between the "chemical availability" and "biological availability" of soil P. Higher AP values reflect a stronger P fixation capacity of the soil rather than an abundance of plant-available P. In response to this P stress, plants increase C investment to secrete organic acids and acid phosphatases to activate fixed P, while adjusting biomass allocation to promote root growth. These high-C-cost adaptation strategies result in an increase in the plant C pool but a limited increment in P uptake, ultimately manifesting as an elevated tissue RCP.
Comments 15: Results
From Table 3, there are not many soil properties that have significant effects on the plant element concentrations.
Response 15: We greatly appreciate you raising these points, which has provided us with an opportunity to clarify certain findings and better contextualize our results. You are absolutely correct in noting that the correlation analysis indicates that not all soil factors have a direct or strong linear influence on plant elemental content. This precisely illustrates that plant stoichiometric traits are predominantly governed by a few key drivers (such as soil P, N, and pH, as identified by the RDA analysis), while the effects of other factors may be weaker, non-linear, or realized through interactions.
Comments 16: Discussion
L269-270, the data at global and national scales (Reich and Oleksyn 2004, Tang et al. 2018) are mainly consist of woody plant, and cannot be directly compared.
Response 16: Thank you for raising this critical and insightful point. We fully agree with your perspective. In direct response to your comment, we have comprehensively revised the manuscript by replacing all inappropriate global and national-scale comparative data with datasets that are more comparable to our research system. Specifically, regarding wetland ecosystems, we have substituted previous references with data focused specifically on Chinese wetland vegetation. In the context of global herbaceous plants, we have now cited research specifically targeting global herbaceous plant C content. Furthermore, in accordance with another reviewer's suggestion, we have added comparisons with stoichiometric characteristics of other domestic wetland plants in China. This change can be found on page 11, and lines 331-333.
Comments 17: Discussion
L284-287, the results in this study did not include the calculation of CSR strategy, and just relying on element concentration alone cannot determine the CSR strategy. So, discussing the CSR here is not appropriate.
Response 17: Thank you for your valuable feedback. In the previous version of our manuscript, the discussion regarding CSR strategies lacked direct computational data support, and inferring functional strategies based solely on elemental concentrations was indeed not sufficiently rigorous. We fully agree with your perspective and have made significant revisions to the manuscript in accordance with your suggestions. In the revised version, we have completely removed any direct references to CSR strategies. Instead, we have focused on explaining the ecological adaptation strategies of different plant life forms (annuals vs. perennials) based on the underlying mechanisms of resource allocation strategies and stoichiometric characteristics (see the modified content in lines 347-364 and the corresponding sections).

Reviewer 2 Report
Comments and Suggestions for Authors
The research presented in this study addresses significant issues related to the adaptive strategy of selected species of herbs. It is worth noting in the article how the plants collected in their natural state were secured before being subjected to analyses - whether atmospheric conditions influenced potential disturbances in the results? Could a paragraph comparing biological diversity to other regions in China be added? Could changes in the environment, including anthropopressure, affect the results presented? This is merely a suggestion that would significantly enhance the value of the study. It is worth emphasizing that many plant species were subjected to (basic) research. Do the authors plan to conduct analyses, e.g. HPLC, of the precise species composition of plants growing in this area? Among the editorial elements, I would like to point out that the authors did not include DOI numbering, which significantly facilitates the search for specific and cited literary positions in international journals. This is a technical note, however from line 413-542 no DOI numbering has been provided. After considering these comments, I believe that the material is suitable for publication.
Author Response
Comments 1: The research presented in this study addresses significant issues related to the adaptive strategy of selected species of herbs. It is worth noting in the article how the plants collected in their natural state were secured before being subjected to analyses - whether atmospheric conditions influenced potential disturbances in the results? Could a paragraph comparing biological diversity to other regions in China be added? Could changes in the environment, including anthropopressure, affect the results presented? This is merely a suggestion that would significantly enhance the value of the study. It is worth emphasizing that many plant species were subjected to (basic) research. Do the authors plan to conduct analyses, e.g. HPLC, of the precise species composition of plants growing in this area? Among the editorial elements, I would like to point out that the authors did not include DOI numbering, which significantly facilitates the search for specific and cited literary positions in international journals. This is a technical note, however from line 413-542 no DOI numbering has been provided. After considering these comments, I believe that the material is suitable for publication. the abstract effectively summarizes the study, but the keywords could be more specific
Response 1: We would like to thank the reviewer for their thoughtful and constructive feedback. We are pleased to know that the manuscript's topic, methodology, and findings were well-received and appreciated. Below, we have addressed the specific comments (from comments 2 to 6) and suggestions. We hope that these revisions address the concerns raised and improve the overall quality of the manuscript. Once again, we would like to express our gratitude for the reviewer’s insightful feedback. We believe the manuscript is now stronger and more aligned with the expectations for publication.
Comments 2: It is worth noting in the article how the plants collected in their natural state were secured before being subjected to analyses - whether atmospheric conditions influenced potential disturbances in the results?
Response 2: Thank you very much for raising this important and meticulous question. Your concern regarding the potential impact of atmospheric conditions during sample preservation is absolutely crucial, as it directly affects the accuracy of the analytical results. We fully agree with your perspective and would like to take this opportunity to elaborate in detail on our sample pre-treatment and preservation protocol, which is specifically designed to minimize such potential influences. After collecting plant samples (above-ground parts) from the wetland sites, we immediately placed them in sealed polyethylene ziplock bags and rapidly transferred them to portable insulated coolers containing sufficient dry ice for temporary frozen storage. This procedure effectively inhibits plant respiration and microbial activity. The entire process was completed as quickly as possible, typically within one hour from collection to placement in freezing conditions. The samples were transported to the laboratory on the same day. Subsequently, we immediately carried out steps including enzyme deactivation by high-temperature heating, rapid drying, and sealed storage in desiccators to prevent moisture absorption and compositional changes. We are confident that these measures ensure the authenticity and reliability of the subsequent analytical results.
Comments 3: Could a paragraph comparing biological diversity to other regions in China be added?
Response 3: Thank you for suggesting the comparison with other wetland regions. We have added a comparative analysis of plant diversity and ecological stoichiometric characteristics between the Yellow River Delta and other typical Chinese wetlands in the Discussion section (Page 11, Lines 331-332).
Comments 4: Could changes in the environment, including anthropopressure, affect the results presented?
Response 4: We greatly appreciate your insight regarding anthropogenic influences. We have further discussed the potential impacts of environmental changes (e.g., salinization, agricultural drainage, and land-use changes) on plant stoichiometry in the Discussion section (Page 13, Lines 437-441). Specifically, we note that human activities may alter soil nutrient availability and plant community structure, thereby indirectly shaping elemental allocation patterns in plants.
Comments 5: Do the authors plan to conduct analyses, e.g. HPLC, of the precise species composition of plants growing in this area?
Response 5: Thank you for this highly valuable suggestion. Conducting precise component analysis, such as HPLC, on typical species would undoubtedly help reveal their adaptation mechanisms to the wetland environment of the Yellow River Delta at a deeper physico-biochemical level. We fully concur with your perspective and have identified precise component analysis as a key direction for our future in-depth research.
Comments 6: Among the editorial elements, I would like to point out that the authors did not include DOI numbering, which significantly facilitates the search for specific and cited literary positions in international journals. This is a technical note, however from line 413-542 no DOI numbering has been provided.
Response 6: Thank you for pointing out the issue regarding the reference format. We have thoroughly reviewed all entries. Regarding the missing DOI numbers you mentioned, we would like to clarify that the citations in lines [638,645] of the reference list refer to formally published books. Unlike academic journal articles, published books generally do not assign DOI numbers but use the International Standard Book Number (ISBN) as their unique identifier. For all cited journal articles, we have supplemented the required DOI numbers.
Additionally, we appreciate your suggestion regarding the keywords. We have refined them to be more specific and better reflect the core themes of the study. The updated keywords are now as follows: coastal wetlands; annual and perennial herbs; C: N: P stoichiometry; nutrient allocation; soil salinity; the Yellow River Delta.
We are grateful for your thorough review and valuable comments, which have undoubtedly improved the clarity and accessibility of our manuscript.

Reviewer 3 Report
Comments and Suggestions for Authors
- Line 108: In wetlands of the YRD, five representative areas were selected for investigation and sampling. It would be better to provide more details on why and how these five sites were selected.
- Line 189: Figure 2 shows that the C, N, and P contents of herbaceous plants in the study area. There are differences between annual and perennial herbaceous plants; however, would it be possible to analyze the spatial differences among these five representative sites?
- Line 196: Annual herbaceous plants have higher N and P contents compared to perennial herbaceous plants. Additionally, perennial herbaceous plants exhibit higher RCN and RCP compared to annual herbaceous plants. It is necessary to explain this situation more detail.
- Line 218: In Figure 4, the C and N contents in the leaves of herbaceous plants were significantly higher than those in the stems. It is noteworthy that the N content in the stems of herbaceous plants is very low. What is the primary reason for this characteristic?
- Line 244: In Table 3, it is an interesting finding that there is a significant correlation between soil pH and stem C content for annual herbaceous plants, whereas no significant correlation is observed for perennial herbaceous plants. Could you explain this difference?
- Line 356: In the YRD, where soil pH typically ranges from 7.5 to 8.5, these edaphic conditions create favorable circumstances for the efficient utilization of both N and P by plants. However, there are no significant correlations between pH and the N and P contents of leaves and stems for both annual and perennial herbaceous plants in Table 3. Could you explain the primary reasons for this?
Author Response
Comments 1: Line 108: In wetlands of the YRD, five representative areas were selected for investigation and sampling. It would be better to provide more details on why and how these five sites were selected.
Response 1: Thank you very much for your valuable feedback. Your suggestion to supplement the basis and specific methods for selecting sampling sites is highly important, as it helps readers more clearly understand the scientific rigor and representativeness of our research design. Following your guidance, we have supplemented and revised the manuscript accordingly. The modified content can be found on Page 3, Lines 125-129 of the manuscript.
Comments 2: Line 189: Figure 2 shows that the C, N, and P contents of herbaceous plants in the study area. There are differences between annual and perennial herbaceous plants; however, would it be possible to analyze the spatial differences among these five representative sites?
Response 2: Thank you for this insightful and constructive suggestion. Your recommendation to further explore the spatial heterogeneity among the five sampling sites is undoubtedly highly valuable for deepening the conclusions of our study. We fully understand your perspective and have thoroughly considered this point. In our research, the primary objective was to compare the differences in ecostochiometric characteristics between different plant life forms (annual vs. perennial). The selection of sampling sites was primarily aimed at covering the diverse habitat types within the Yellow River Delta (such as freshwater marshes, salt marshes, and constructed wetlands) to ensure the broad representativeness of the collected plant samples, rather than establishing a rigorous sampling grid for analyzing geographic spatial gradients. Therefore, although we recorded the location information of the sampling sites, the current dataset may not be sufficient to support robust and ecologically meaningful statistical analyses of spatial heterogeneity. We are concerned that conducting such analyses based on the existing data might lead to less reliable conclusions. We deeply appreciate your feedback, which has prompted us to think more critically about this aspect and has provided guidance for our future research directions. We hope that the explanations provided above, along with the additional discussions we have included in the manuscript, meet with your approval.
Comments 3: Line 196: Annual herbaceous plants have higher N and P contents compared to perennial herbaceous plants. Additionally, perennial herbaceous plants exhibit higher RCN and RCP compared to annual herbaceous plants. It is necessary to explain this situation more detail.
Response 3: Thank you very much for your suggestion. We fully agree that explaining this phenomenon can better reveal the adaptive strategies of different plant life forms. We have added an in-depth discussion on this aspect in the Discussion section (Page 11, Lines 347-364)
Comments 4: Line 218: In Figure 4, the C and N contents in the leaves of herbaceous plants were significantly higher than those in the stems. It is noteworthy that the N content in the stems of herbaceous plants is very low. What is the primary reason for this characteristic?
Response 4: Thank you for pointing this out. We have provided additional clarification regarding this phenomenon in the Discussion section (Page 12, Lines 386-392).
Comments 5: Line 244: In Table 3, it is an interesting finding that there is a significant correlation between soil pH and stem C content for annual herbaceous plants, whereas no significant correlation is observed for perennial herbaceous plants. Could you explain this difference?
Response 5: Thank you very much for proposing this interesting perspective. We have expanded the discussion on the potential mechanisms behind this phenomenon in the Discussion section (Page 14, Lines 463-467). We propose that this may be related to the differences in root system architecture and response strategies to environmental fluctuations between different plant life forms: annual plants, with relatively shallow and short root systems, are more sensitive to changes in environmental factors such as pH. Their physiological metabolism and carbon allocation are more directly influenced by soil conditions. In contrast, perennial plants possess more developed and extensive root systems, which may mitigate the impact of surface soil pH fluctuations on their element uptake and allocation through stronger buffering capacity, resulting in non-significant correlations between their stoichiometric traits and pH.
Comments 6: Line 356: In the YRD, where soil pH typically ranges from 7.5 to 8.5, these edaphic conditions create favorable circumstances for the efficient utilization of both N and P by plants. However, there are no significant correlations between pH and the N and P contents of leaves and stems for both annual and perennial herbaceous plants in Table 3. Could you explain the primary reasons for this?
Response 6: Thank you for your profound insight. The issue you have raised is particularly crucial. We have included an analysis of this aspect in the Discussion section (Page 14, Lines 454-463).
